# Local Anesthetic Delivered with a Dual Action Ring and Injection Applicator Reduces the Acute Pain Response of Lambs during Tail Docking

**DOI:** 10.3390/ani11082242

**Published:** 2021-07-30

**Authors:** Alison Small, Danila Marini, Ian Colditz

**Affiliations:** CSIRO Agriculture and Food, Armidale, NSW 2350, Australia; Danila.marini@csiro.au (D.M.); ian.colditz@csiro.au (I.C.)

**Keywords:** analgesia, sheep, ischemia, latex ring, rubber ring, de-tailing, behavior

## Abstract

**Simple Summary:**

Tail docking is a procedure practiced on millions of lambs all over the world. The objective is to prevent fecal soiling on the lower part of the tail, reduce soiling of the breech, and thereby lessen the risk of blowfly strike. Docking can be done with a knife or a clamp, but applying a latex ring round the tail, cutting off the blood supply so that the tail drops off a few weeks later, is the most popular method. All methods cause acute pain which diminishes substantially after the first hour. The present trial determined whether local anesthetic delivered by a prototype Numnuts^®^ device, a novel, dual-function applicator, would reduce this pain in two to four-week-old lambs. Comparison of lambs that were sham handled, lambs that underwent ring tail docking and a third group of lambs that underwent ring tail docking and that were injected with lignocaine using the dual function device was conducted. All lambs were returned to their pen with their mothers and videoed for three hours for behavioral signs of pain. Every five minutes for the first hour and then every ten minutes, each lamb’s posture, movement and feeding behavior was classified and quantified, and the data subjected to statistical analysis. It was concluded that applying lignocaine using the novel device greatly reduced the degree of pain observed.

**Abstract:**

Docking the tail of lambs is a standard husbandry procedure and is achieved through several techniques including clamps, hot or cold knives and latex rings, the last of which is the most popular. All tail docking methods cause acute pain which can be reduced by application of local anesthetic, however precise anatomical injection for optimal efficacy requires considerable skill. This pen trial evaluated the ability of local anesthetic (LA) delivered with a dual function ring applicator/injector to alleviate acute tail docking pain. Thirty ewe lambs were assigned to one of three treatment groups (*n* = 10 per group): ring plus local anesthetic (Ring LA), ring only (Ring) and sham handled control (Sham). Lambs were videoed and their behavior categorized every five minutes for the first hour and every 10 min for the subsequent two hours after treatment. There was a significant effect (*p* < 0.001) of treatment on total active pain related behaviors in the first hour, with Ring lambs showing higher counts compared to Ring LA or Sham. Ring lambs also displayed a significantly higher count of combined abnormal postures (*p* < 0.001) than Ring LA or Sham lambs. Delivery of 1.5 mL of 2% lignocaine via the dual action device abolished abnormal behaviors and signs of pain in Ring LA lambs. However, lambs in the Ring LA group spent less time attempting to suckle compared to Ring and Sham lambs, suggesting that some residual discomfort remained.

## 1. Introduction

Docking the tails of lambs by applying a vasoconstrictive latex rubber ring is a widespread practice used in many countries. The procedure causes acute pain and stress that lasts for over an hour [1,2,3]. Trials have shown that when local anesthetic (LA) is injected into the tail prior to the ring being applied, pain can be greatly alleviated [1]. The vast majority of farmers would prefer to cause as little pain as possible when docking their animals [4], however injection of LA is slow and cumbersome, and requires considerable technical skill for accurate location of the injection site [5]. To administer LA and a rubber ring currently requires two tools: a syringe and needle to inject the LA, and a set of marking pliers to fit a ring on the tail. Furthermore, injection with a syringe and needle risks needle stick injury. It is therefore unsurprising that very few commercial-scale farmers use local anesthetic for tail docking.

To address these issues of logistics, ergonomics and operator safety, a novel dual function marking instrument (Numnuts^®^) was developed by Senesino Ltd., (Glasgow, UK), that allows the operator to fit a latex docking ring around a lamb’s tail and then inject local anesthetic into the tail adjacent to the ring (www.numnuts.store/the-development/, accessed on: 17 July 2021). The Numnuts^®^ device provides accurate and consistent local anesthetic application without a requirement for detailed knowledge of animal anatomy or extensive operator training. The trial described here examined the degree of pain relief provided by docking the tail of lambs using a late-stage prototype of the Numnuts^®^ device. 

## 2. Materials and Methods

The animal phase of this study was carried out at the Moredun Research Institute, Midlothian Scotland, under UK Home Office License PPL 70/8075, and the data analysis was carried out at the Commonwealth Scientific and Industrial Research Organization (CSIRO), Armidale New South Wales, Australia.

### 2.1. The Applicator Device 

A protype version of a novel dual-function marking tool was developed. The device allows the operator to rapidly fit a rubber ring around a lamb’s tail and then inject 1.5 mL of local anesthetic past the ring into the tail, just cranial (proximal) to the constriction site. When the rubber ring contracts over the tail, it temporarily holds the prongs of the device in a fixed position around the tail. This temporary fixation enables the injection mechanism to consistently deliver a metered 1.5 mL volume of local anesthetic subcutaneously into the tissue of the tail beneath the ring.

### 2.2. Design of the Efficacy Trial

The efficacy of local anesthetic (1.5 mL 2% lignocaine hydrochloride, Troy laboratories, Australia) injected using a late-stage prototype of the Numnuts^®^ applicator (Senesino Ltd., Glasgow, UK) was examined in 2 to 4-week old, Greyface cross Texel ewe lambs. Thirty single-born lambs (5.8–11.8 kg) were assigned to three treatment groups (*n* = 10 per group): ring plus local anesthetic (Ring LA), ring only (Ring) and sham handled control (Sham) groups were balanced by stratified randomization on weight. 

Lambs were individually identified by large colored numerals sprayed on their flanks. The lambs were housed as ewe-lamb pairs in group pens (8 × 5 m) with deep straw bedding over concrete floors at the Moredun Research Institute, Bush Loan, Edinburgh, Scotland (Figure 1). Pens housed 7–10 lambs each. Activity in each pen was recorded by two video cameras connected to digital video recorders positioned on opposite sides of the pen, and footage was captured by a video management software (Huawei Technologies Co., Ltd., Reading, UK). 

Treatment application took 1 min to undertake. During treatment application, lambs were restrained in dorsal recumbency in a marking cradle. Ring lambs had rubber rings (Elastrator Brand) applied using the prototype applicator without an injection, Ring LA received an injection of 1.5 mL lignocaine via the prototype applicator at the time of ring application. Sham controls had their tail manipulated without application of a ring or injection. After the procedure, the lambs were returned to their pens, with each pen containing a mix of treatment groups. Due to operator error, although all treatments were represented in each pen, treatments were not fully balanced between pens.

The responses of the lambs were videoed for analysis of active pain avoidance behaviors. Behaviors and descriptions used in this study (Table 1 and Table 2) have been previously validated as pain related behaviors in response to ring castration and tail docking [6,7]. Personnel quantifying the behaviors were blinded for the treatment group, having not been present at treatment administration. Video resolution was not sufficient to identify which lambs had rings and which did not. Postures were classified and scored at 5-min intervals for the first hour and at 10-min intervals for the second and third hours, as shown. Active pain related behaviors were classified every 5 min for the first hour and were summed to give a total count. The time that behaviors were recorded was based on the lamb’s treatment time, as time zero for each individual. Teat seeking behavior was also classified during the scoring of active behaviors.

### 2.3. Analysis of the Results

Counts for postures were summed over two intervals: the first hour, and hours 2 and 3 combined, so that 12 counts were recorded per animal for hour 1 and 12 counts for hours 2 and 3 combined. In hours 2 and 3, the number of observations for each lamb varied between 7 and 12. In this interval, some observations were missed when a lamb was obscured by other animals in the pen. For animals available for observation on fewer than 12 occasions, scores for each posture were rescaled to 12. Analysis of variance (ANOVA) was performed on total counts in the first hour, and a repeated measures model was fitted to examine the change in a variable between the two time blocks, hour 1 and hour 2 and 3. Residuals from the repeated measures models could not be normalized by data transformation. Data for the first hour were suitable for analysis without transformation. Liveweight was tested as a covariate and fitted when significant (*p* < 0.050). Sham handled lambs were not present in all pens, so the pen (i.e. group) was not fitted in the analysis as the pen was confounded with treatment. Data for hours 2 and 3 were not normalized by transformation, and were analyzed by a Kolmogorov–Smirnov non-parametric test. 

Active pain related behaviors were analyzed two ways. For each animal, total active pain behaviors (Table 1) were summed at each of the 12 observation points during the first hour and the change over time was analyzed in a repeated measures model. Data was log transformed for analysis. Secondly, total active pain behaviors for each animal were summed across the 12 observation points and analyzed by ANOVA. Total count of teat seeking activity, eating at the trough and sucking in the first hour, and sucking in hours 2 and 3 were analyzed by ANOVA. Eating at the trough in the first hour was log transformed for analysis. Post-hoc contrasts between treatments were performed by univariate F tests. Eating at the trough in hours 2 and 3 was analyzed by a Kolmogorov–Smirnov non-parametric test. 

Analyses were performed in Systat version 9. Plotted data are least squares means ± standard error, except for back transformed values where error bars are not plotted.

## 3. Results

### 3.1. Behavioral Postures

There was an effect of treatment on the combined count of total abnormal postures in the first hour (*p* < 0.001, Figure 2b) Ring lambs displayed a significantly higher count of combined abnormal postures than Ring LA or Sham lambs (*p* < 0.001). There was no difference in abnormal postures between Sham and Ring LA lambs (*p* = 0.716). Data could not be transformed to normality for repeated measures analysis of change in total abnormal postures between time intervals. In hours two and three, there was a low count of combined abnormal postures in all treatment groups (Figure 2a).

Subsequent analyses on individual types of abnormal postures examined the two time periods independently. For the first hour, there was an effect of treatment on abnormal walking (*p* < 0.002). Ring lambs displayed higher counts of abnormal walking than Ring LA or Sham in the first hour (*p* < 0.002, Figure 2c). There was no difference in abnormal walking between Sham and Ring LA lambs (*p* = 0.763, Figure 2c). There was an effect of treatment on abnormal lying (*p* < 0.001). Ring treatment displayed higher counts of abnormal lying than Ring LA or Sham (*p* = 0.001). There was no difference in abnormal lying between Sham and Ring LA lambs (*p* = 1.000) (Figure 2d). There were few instances of abnormal standing in the first hour and an effect of treatment was not observed (*p* = 0.293, Figure 2e). For hours two and three, there were no differences between treatments for total abnormal postures or for the individual postures of abnormal standing, abnormal walking or abnormal standing.

### 3.2. Active Pain Behaviors in the First Hour, Eating and Sucking 

There was an effect of treatment (*p* < 0.001), time (*p* < 0.001), and a treatment by time interaction (*p* < 0.001) on total active pain behaviors in the first hour. The count of active pain behaviors was higher in Ring lambs than Ring LA lambs at 5, 15, 25 and 30 min (Figure 3a) and approached significance at 10 and 20 min. There was no difference in active pain behaviors between sham and Ring LA lambs at any time point. For total active behaviors summed across the 12 time points, the effect of treatment was highly significant (*p* < 0.001, Figure 3b). Ring lambs displayed a higher count of active pain-related behaviors (*p* < 0.001) than Ring LA or Sham lambs. There was no difference in active pain behaviors between Sham and Ring LA lambs (*p* = 0.861).

There was an effect of treatment on teat seeking (*p* = 0.045). Applying the ring tended to reduce teat seeking behavior in Ring lambs (*p* = 0.08), and decreased teat seeking in Ring LA lambs (*p* = 0.016, Figure 3c), however Ring and Ring LA lambs did not differ (*p* = 0.465). Treatment tended to reduce sucking behavior in the first hour (*p* = 0.054, Figure 3d). There was no effect of treatment on eating at the trough (*p* = 0.191) in the first hour, but the count for this activity was very low (Figure 3e). There was no effect of treatment on eating at the trough or sucking in hours two and three.

## 4. Discussion

The aim of the current study was to look at the efficacy of a novel dual-function marking tool and its ability to reliably provide a single dose of local anesthetic at the time of tail-docking. The current trial showed that injection of 1.5 mL local anesthetic into the tail using the prototype device at the time of ring application abolished abnormal behaviors and signs of pain in the first hour after tail docking. However, the lambs in the Ring LA group spent less time attempting to suckle, suggesting that some residual discomfort may have remained. These results corroborate those of a subsequent field trial, where using the Numnuts**^®^** device to apply the ring and lignocaine suppressed the degree of pain observed following tail docking [8], and align with several other studies which showed the benefits of local anesthetic delivered by syringe and needle [1,9,10].

In this study the application of the local anesthetic lignocaine using a dual function marking tool was able to reduce the acute pain experienced by lambs undergoing ring tail docking. Lignocaine has been reported to provide consistent pain relief to lambs in the first hour following ring castration and tail docking, with reduced cortisol response and reduced display of pain related behaviors in lambs receiving lignocaine following tail docking compared to lambs that did not receive lignocaine [11,12,13]. Although pain related behaviors were reduced in the lambs that received lignocaine in the current study, lambs in this group still had reduced teat seeking and sucking behavior, indicating some residual pain. 

As lignocaine is a fast-acting short duration analgesic, only the acute pain phase in response to tail-docking was examined. There has been extensive research into the acute and chronic effects of ring tail-docking and castration [7,14,15,16,17]. The acute pain phase of ring tail-docking has been reported to last up to an hour [18], with measures of chronic pain in lambs being hard to observe and unvalidated making it difficult to determine differences in the long term. Previous work has found no difference in chronic pain responses or in growth rate in lambs castrated and tail-docked with rubber rings that were applied with and without local anesthetic [13]. However, prevention of secondary hyperalgesia has been reported in lambs receiving local anesthetic following ring castration and tail docking compared to those without [19]. Long term effects were not looked at in this study, as the effects of local anesthetic at relieving tail docking pain are well reported and scope of this study was to look at the device’s efficacy at providing a single dose of local anesthetic. As most differences in treatment (with and without local anesthetic) following ring castration and tail docking are observed in the acute pain phase [13,18,19], only this phase was examined in the current study.

The provision of pain relief for lambs undergoing painful husbandry procedures has increased over the last few years. Producers now have access to registered, easy to use products such as Tri-Solfen [20,21,22] and Buccalgesic [23,24,25] for surgical procedures such as mulesing, castration and tail-docking. Producers have faced limitations in their ability to provide feasible acute pain relief provision for ring tail docking [8]. Previous research has looked at the use of Tri-Solfen in providing pain relief for lambs undergoing hot knife tail docking (the lambs were concurrently ring castrated without the application of LA), however the topical formulation had minimal impact on behaviors when applied to the open wound on the tail in lambs that were concurrently castrated with a ring [26]. There has also been work that has looked at coating rubber rings in lignocaine as a method for delivering pain relief [9]. The lignocaine-coated rings ameliorated some of the pain in response to ring castration when compared to normal rings, however they were not as effective as injections of lignocaine [9]. Absorption of lignocaine through intact skin is limited, and delivery of pain relief via this route is slow and does not adequately address the acute pain phase of ring castration and tail docking. In the present study, application of a metered dose of lignocaine using the prototype Numnuts® device significantly reduced the acute pain response in lambs that underwent ring tail docking. 

## 5. Conclusions 

The present study demonstrated that a novel dual-function marking tool can provide a measured 1.5 mL dose of lignocaine to the site of tail docking. This led to immediate pain relief for the acute pain phase of ring tail docking in lambs that received the local anesthetic compared to those that were not provided with pain relief. The present results are a positive step towards providing sheep producers with a device that would allow a rapid, practical and safe method for providing large scale relief of pain caused by tail docking. 

## Figures and Tables

**Figure 1 animals-11-02242-f001:**
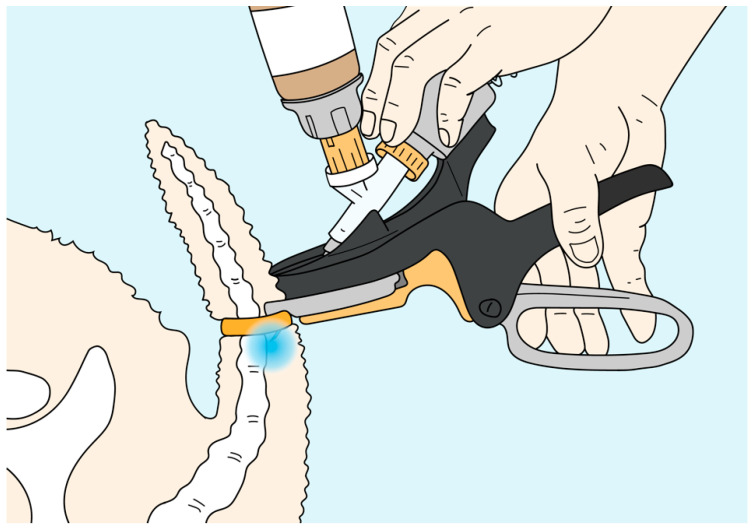
The protype used in the study was functionally identical to the stylized version illustrated here, which includes design modifications made to enable commercial manufacturing of the device. The diagram shows where local anesthetic is injected, proximal to the ring, when using the Numnuts^®^ device.

**Figure 2 animals-11-02242-f002:**
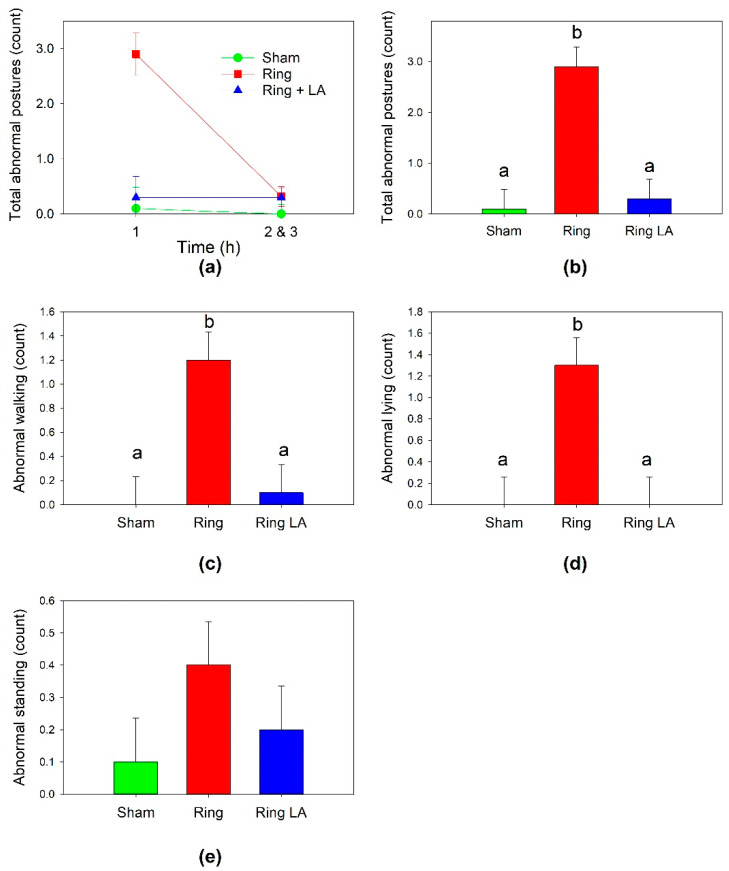
Abnormal postural behaviors in lambs after tail docking with (Ring LA) or without (Ring) anesthetic, or sham treatment (Sham). Individual graphs show lambs’ (**a**) total abnormal postures in hour 1, and hours 2 and 3, (**b**) total abnormal postures in the first hour, (**c**) abnormal walking in the first hour, (**d**) abnormal lying in the first hour (**e**) abnormal standing in the first hour. Data points with a matching lowercase letter indicate no significant differences between those data points.

**Figure 3 animals-11-02242-f003:**
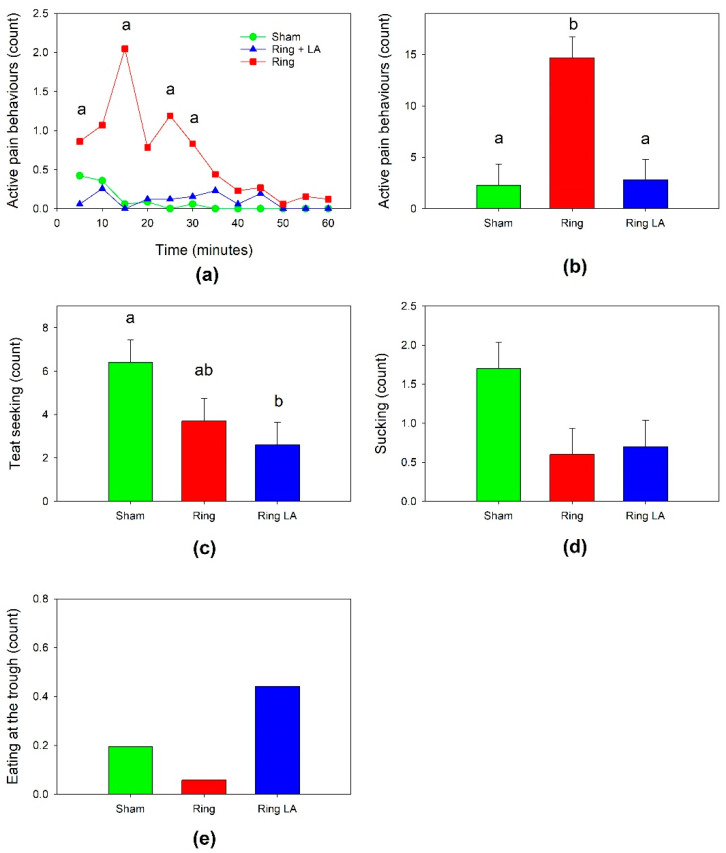
Active pain avoidance behaviors in lambs after tail docking with (Ring LA) or without (Ring) anesthetic, or sham treatment (Sham) in the first hour. (**a**) Count of total active pain behaviors over time (minutes), (**b**) total active pain behaviors (counts over entire duration), (**c**) teat seeking (counts over entire duration), (**d**) sucking (counts over entire duration), and (**e**) eating at the trough (counts over entire duration). Data points with a matching lowercase letter indicate no significant differences between those data points. See text for *p* values.

**Table 1 animals-11-02242-t001:** Descriptions of active pain avoidance behaviors recorded during the experiment.

Behavior	Abbreviation	Description
Restlessness	Rst	Transition from standing to lying or vice versa within a 30 s window at the observation timepoint
Kicking/foot stamping	FSK	Either a front or hind limb (usually hind limb) was lifted and forcefully placed on the ground while standing or was used to kick while standing or lying
Rolling	rl	Lamb rolled from lying on one side to the other without getting up or rolled on its back and then returned to lying on the same side.
Jumping	jmp	Lamb moved forward using bunny hops with its hind limbs
Licking/biting wound site	LBW	Movement of the head beyond the shoulder, including both looking and touching at the source of pain and grooming.
Easing quarters	EQ	Abnormally lowers rear quarters (standing) or attempts to keep quarters off the ground (lying).
Teat seeking	TS	No differentiation with or without sucking
Pain behaviors	Rst + FSK + rl + jmp + LBW + EQ	All pain avoidance behaviors pooled

**Table 2 animals-11-02242-t002:** Descriptions of postural behaviors recorded during the experiment.

Behavior	Abbreviation	Description
Normal ventral lying	V1	Lying on sternum with legs tucked in and head up or down
Abnormal ventral lying	V2	Ventral lying with hind limbs partially or fully extended or keeping scrotal region off the ground (dog sitting)
Ventral lying other	Vu	Lamb was lying ventrally but unable to clearly categorise the lying posture
Lateral lying	L	Lateral (on side) with one shoulder on ground, extension of hind limbs with head up or down
Abnormal lying	L + V2	Abnormal lying categories pooled
Total lying	V1 + V2 + Vu + L	All lying categories pooled
Normal standing	S1	Standing with no apparent abnormalities
Statue standing	SS	Immobile standing with an obvious withdrawal from interaction with other pen members and outside stimuli. Legs positioned further back than normal. Can show arched back
Abnormal standing	S2	Standing hunched or unsteadily, often associated with foot stamping, kicking and tail wagging
Standing other	Su	Lamb was standing but unable to clearly categorise the standing posture
Normal walking	W1	Walking with no apparent abnormalities
Abnormal walking	W2	Walking unsteadily or stiffly, includes walking backwards, on knees, moving forward with bunny hops, circling, leaning or falling.
Walking other	Wu	Lamb was walking but unable to clearly categorise the walking type
Feeding	Feed	Feeding at the trough
Sucking	Sk	Drinking from the ewe
Total standing	S1 + S2 + SS + Su + W1 + W2 + Wu	All standing and walking categories pooled
Total Abnormal postures	V2 + SS + S2 + W2 + L	All abnormal posture categories pooled

## Data Availability

The data presented in this study are available on request from the corresponding author.

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
