# Peer review of "Local Anesthetic Delivered with a Dual Action Ring and Injection Applicator Reduces the Acute Pain Response of Lambs during Tail Docking"

_animals, 2021, doi:10.3390/ani11082242_

Round 1
Reviewer 1 Report
This is an important study demonstrating the efficacy of a novel device for providing subcutaneous lignocaine to lambs being docked using a rubber ring. Ischaemic pain is difficult to ameliorate without lignocaine infiltration, so the results from this study will have great impact on the welfare of lambs undergoing this procedure.
Overall the study is well described, particularly the methodology and results. The discussion however is limiting. There is almost no reference to the specific results. It would be useful to include some discussion regarding the behaviours observed, in reference to the literature. The suckling data presents an interesting trend that could be discussed. Why do you think that despite displaying no pain behaviours, Ring LA lambs still had reduced teat seeking behaviour?
I think these additions are only minor, and with some further development of the discussion, the manuscript should be published.
Author Response
General comments:
This is an important study demonstrating the efficacy of a novel device for providing subcutaneous lignocaine to lambs being docked using a rubber ring. Ischaemic pain is difficult to ameliorate without lignocaine infiltration, so the results from this study will have great impact on the welfare of lambs undergoing this procedure.
Overall the study is well described, particularly the methodology and results. The discussion however is limiting. There is almost no reference to the specific results. It would be useful to include some discussion regarding the behaviours observed, in reference to the literature. The suckling data presents an interesting trend that could be discussed. Why do you think that despite displaying no pain behaviours, Ring LA lambs still had reduced teat seeking behaviour?
We believe that discussing this finding in detail is beyond the scope of the current manuscript, as it was not a primary focus of the study. Reduced teat seeking behaviour could be indicative of a continued effect on appetence or affect but further work would be required to confirm or refute this.
I think these additions are only minor, and with some further development of the discussion, the manuscript should be published.
4) The methods and materials need some additional information. Was the selection of behaviours in the ethogram based on [6]?
L95: Behaviors and descriptions used in this study have been previously validated as pain related behaviors in response to ring castration and tail docking [6, 7]
5) How long did it take to apply the treatments?
L 110: Treatment application took 1 minute to undertake.
6) Were lamb behaviours recorded for each lamb at its own interval post-treatment or for all lambs at an average duration post-treatment?
L135: The time that behaviors were recorded was based on the lamb’s treatment time as time zero.
7) The analysis of results section requires some additional information such as the statistical software used and the specific statistical test used. Did the authors consider using non-parametric tests for variables that could not be normalised?
L149 – 157 and L 167 – 174: Details of statistical software have been added. Data for postures could not be transformed to normalise residuals from repeated measures models. We are unaware on a non-parametric method for examining treatment by time interaction in a repeated measures design.
193-195 Results of non-parametric analysis of abnormal postures in H2&3 are now reported.
8) I would also like to know how the authors dealt with 0 values (particularly in the analysis of time point data as per fig 3a). What was the rationale for rescaling the results of lambs that had fewer than 12 observations? How was the rescaling conducted?
The repeated measures model for data presented in figure 3a accepted zero values. Counts were rescaled so that all lambs had the equivalent of 12 observations. Say 10 of the planned 12 observation were achieved for a lamb and at 2 of those observations times the lamb was standing. The count for that lamb for standing was rescaled as follows: 2/10 = x/12, solving for x, x = (2 * 12) / 10 = 2.4. Observations were only missed when scoring postures in hours 2 and 3. As noted above, we ended up not analysing abnormal postures in these timepoints due to the low count of abnormal postures (figure 2a) and inability to transform data to normalise residuals from repeated measures models.
9) Throughout the results section the authors include the word “significant” which is redundant if p-values are reported. Please remove.
The term “significant” has been removed and the section has been edited.
10) Throughout the results the authors report results as “Ring displayed” it is not the ring that displayed the behaviour but the lamb that was in the Ring treatment. Please revise.
Revised.
11) Finally, the graphs supplied in the manuscript are far to small to be able to provide a meaningful image. Please make these larger.
Graphs have been re-drawn.
12) The discussion is short and does not discuss the results that were actually generated in the study. I appreciate that this is a small study but the discussion really only covers the methods used to apply local anesthetic to lambs during the tail docking procedure.
The discussion has been edited a new paragraph has been added L 275-294.
13) The conclusion includes factors that were not recorded in the study. Please keep the conclusion to only what was investigated in the study itself. It reads as an advertisement for the device.
The conclusion has been revised.
Specific comments
14) Title: Please revise to “Local anesthetic delivered with a dual action ring and injection applicator reduces the acute pain behaviours of lambs after tail docking”
Revised as suggested.
15) Line 17 – the description of the groups makes it unclear how many there were
L 17-18: Comparison of lambs that were sham handled, lambs that underwent ring tail docking and a third group of lambs that underwent ring tail docking and injected with lignocaine using the dual function device was conducted.
16) Simple summary: it would be valuable to include the age of the lamb here
L16: in 2 to 4-week-old lambs
17) Line 55 to 57 – please provide some evidence for this statement
Details of the development process are given in confidential engineering reports, and summarised on the company website https://numnuts.store/the-development/
18) Line 72 – what was the birth rank of the lambs. This could affect lamb competition for the udder if more than a single lamb was present
The lambs in this study were single-born lambs, so no birth-rank effect would be discernible. The words ‘single-born’ have been added at line 102 in the tracked version.
19) Figure 1 – I am usually in favour of the addition of images to aid understanding, but I don’t think this figure adds to understanding the study design, remove
Figure 1 has been replaced with an image of the device
20) Line 94 – how were observers blinded, surely, they could see the presence of the ring
The resolution of the video was such that it was very difficult to see the ring, and even if the ring was visible, the observer would still be blinded to RING vs RING-LA
21) Table 1 – I suggest this table is split into two. One showing “Active pain avoidance” and the other “Postural behaviours”. Change “suckling” to “sucking” (ewes suckle, lambs suck)
Table has been split into two, suckling changed to sucking.
22) Line 107 – why were sham lambs not present in all pens?
Due to operator error, although all treatments were represented in each pen, treatments were not fully balanced between pens.
23) Line 111 – for what reasons were lambs missed from the observations?
L152 “Lamb observation was missed if they were obscured by other animals in the pen”
24) Line 122 – if data could no be normalised how were the differences determined? This should be clearly stated in the methods
Description of statistical methods has been revised. Data could not be normalised for repeated measures analysis of change in abnormal posture counts from H1 to H 2&3, so this particular analysis was not performed. Other differences reported are from parametric and non-parametric analyses within H1 and Hours 2&3.
25) Figure 2 – please standardise y axis labels (lying abnormally + abnormal lying). I suggest the final sentence of the caption be rephrased as “Frequencies with different lowercase letters indicate....” and include the significance level (P<0.05)
Figures have been redrawn and axis labels standardised
26) Line 150 change reference to figure to (Figure 3a) the “a” does not provide any additional information
Edited
27) Line 162-163 – This sentence is not reporting a result but an interpretation of the results. The variable measured was lamb suckling behaviour not the lamb’s ability to suck from the ewe.
Sentence has been removed
28) Figure 3 – please include the units for (a) in the caption
Included
29) The conclusion covers aspects of the device that are outside the scope of the study. It reads as an advertisement for the device and must be revised to include only the aspects of the device that were investigated in this manuscript.
The conclusion has been revised
Reviewer 2 Report
Reviewer report Animals 1261496
“Local anesthetic delivered with a dual action ring and injection applicator Numnuts® reduces the acute pain response of lambs during tail docking”
This paper describes a study that examined the impact of delivering local anesthetic at the time of application of a rubber ring for the purpose of tail docking lambs. This is a nice concise study but the manuscript requires some revision before it would be acceptable for publication.
General comments
The methods and materials need some additional information. Was the selection of behaviours in the ethogram based on [6]? How long did it take to apply the treatments? Were lamb behaviours recorded for each lamb at its own interval post-treatment or for all lambs at an average duration post-treatment? The analysis of results section requires some additional information such as the statistical software used and the specific statistical test used. Did the authors consider using non-parametric tests for variables that could not be normalised? I would also like to know how the authors dealt with 0 values (particularly in the analysis of time point data as per fig 3a). What was the rationale for rescaling the results of lambs that had fewer than 12 observations? How was the rescaling conducted?
Throughout the results section the authors include the word “significant” which is redundant if p-values are reported. Please remove. Throughout the results the authors report results as “Ring displayed” it is not the ring that displayed the behaviour but the lamb that was in the Ring treatment. Please revise. Finally, the graphs supplied in the manuscript are far to small to be able to provide a meaningful image. Please make these larger.
The discussion is short and does not discuss the results that were actually generated in the study. I appreciate that this is a small study but the discussion really only covers the methods used to apply local anesthetic to lambs during the tail docking procedure.
The conclusion includes factors that were not recorded in the study. Please keep the conclusion to only what was investigated in the study itself. It reads as an advertisement for the device.
Specific comments
Title: Please revise to “Local anesthetic delivered with a dual action ring and injection applicator reduces the acute pain behaviours of lambs after tail docking”
Line 17 – the description of the groups makes it unclear how many there were
Simple summary: it would be valuable to include the age of the lamb here
Line 55 to 57 – please provide some evidence for this statement
Line 72 – what was the birth rank of the lambs. This could affect lamb competition for the udder if more than a single lamb was present
Figure 1 – I am usually in favour of the addition of images to aid understanding but I don’t think this figure adds to understanding the study design, remove
Line 94 – how were observers blinded, surely they could see the presence of the ring
Table 1 – I suggest this table is split into two. One showing “Active pain avoidance” and the other “Postural behaviours”. Change “suckling” to “sucking” (ewes suckle, lambs suck)
Line 107 – why were sham lambs not present in all pens?
Line 111 – for what reasons were lambs missed from the observations?
Line 122 – if data could no be normalised how were the differences determined? This should be clearly stated in the methods
Figure 2 – please standardise y axis labels (lying abnormally + abnormal lying). I suggest the final sentence of the caption be rephrased as “Frequencies with different lowercase letters indicate....” and include the significance level (P<0.05)
Line 150 change reference to figure to (Figure 3a) the “a” does not provide any additional information
Line 162-163 – This sentence is not reporting a result but an interpretation of the results. The variable measured was lamb suckling behaviour not the lamb’s ability to suck from the ewe.
Figure 3 – please include the units for (a) in the caption
Conclusion
The conclusion covers aspects of the device that are outside the scope of the study. It reads as an advertisement for the device and must be revised to include only the aspects of the device that were investigated in this manuscript.
Author Response
30) In the article, it is not mentioned that the study has been approved by an ethical committee. Could the authors clarify this fact? An ethical committee must evaluate this type of studies in which harm is infringed on to animals.
L64 The animal phase of this study was carried out at the Moredun Research Institute, 81 Midlothian Scotland, under UK Home Office Licence PPL 70/8075, and the data analysis was carried out at CSIRO, Armidale NSW Australia.
31) In this trial, only acute pain, which is necessary to control, has been assessed. However, the evaluation of pain responses is challenging, especially in lambs, as they are very stoic animals and do not demonstrate pain as clearly and intensely as other species. This trial only assesses pain during the first three hours after treatment, and it is based on observational measurements such as abnormal behaviors or abnormal postures. This could have been improved with other measurements as serum cortisol or acute-phase proteins. The trial should have also included an assessment of the chronic pain that this type of procedure can generate in the animals due to the inflammation, trauma, stress and secondary infections after tissue necrosis.
The focus of this study was looking at the effects of the dual action device at delivering lignocaine following ring tail-docking. As lignocaine is a fast-acting short duration analgesic only the acute pain phase following ring tail docking was looked at. There has been extensive research into the pain associated with ring tail docking and it is known that the acute pain phase lasts up to an hour (as observed through behavioural change). Generally, in studies looking at pain when blood measures are taken, they are done at 15 minute intervals following treatment application, the process of collecting blood samples introduces interruptions to behaviour and potential pain (associated with needles). Due to this we focused on observing the behaviours alone due to the short observation period, as to not interrupt the lambs behaviour post treatment.
There has also been extensive work looking at long term effects of ring tail docking with or without local anesthetic and have found no difference in growth rate between lambs and some difference in lesions (Kent et al. 2000; Mellema et al. 2006). Most differences are observed in the acute pain phase. Due to the focus of this study being about the effectiveness of the device at administering lignocaine it wasn’t deemed necessary to look at the long term effects.
Kent, J. E., Jackson, R. E., Molony, V., & Hosie, B. D. (2000). Effects of acute pain reduction methods on the chronic inflammatory lesions and behaviour of lambs castrated and tail docked with rubber rings at less than two days of age. Veterinary Journal, 160(1), 33-41. doi:10.1053/tvjl.2000.0465
Mellema, S. C., Doherr, M. G., Wechsler, B., Thueer, S., & Steiner, A. (2006). Influence of local anaesthesia on pain and distress induced by two bloodless castration methods in young lambs. The Veterinary Journal, 172(2), 274-283. doi:https://doi.org/10.1016/j.tvjl.2005.06.002
In response to your question and the editors question 3 a paragraph has been added to the discussion in relation to this L 283-295.
32) The use of rubber rings to perform tail docking and castrations in animals is being banned in more and more countries due to the damage and stress generated to the animal. Although the main objective of this trial is to demonstrate that local anaesthetic can provide pain relief to avoid this damage, it is not considered during the trial the chronic pain, which an anti-inflammatory product could have controlled.
Tail docking is still practiced in many parts of the world and is considered a necessary husbandry procedure to reduce the risk flystrike, particularly in places like Australia. As with the question above the focus of this study was looking at the effects of the dual action device at delivering lignocaine following ring tail-docking. There has been research looking at the effectiveness of non-steroidal anti-inflammatory (such as meloxicam) at providing analgesia to lambs undergoing ring castration and tail docking for long term pain relief. In Australia producers already have access to such pain relieving agents for use during castration and tail docking as well as other options such as flunixin, which can be administered by a veterinarian.
Paull, D. R., Small, A. H., Lee, C., Palladin, P., & Colditz, I. G. (2012). Evaluating a novel analgesic strategy for ring castration of ram lambs. Veterinary Anaesthesia and Analgesia, 39(5), 539-549. doi:10.1111/j.1467-2995.2012.00716.x
Mellor, D. J., & Stafford, K. J. (2000). Acute castration and/or tailing distress and its alleviation in lambs. New Zealand Veterinary Journal, 48(2), 33-43. doi:10.1080/00480169.2000.36156
Small, A. H., Jongman, E. C., Niemeyer, D., Lee, C., & Colditz, I. G. (2020). Efficacy of precisely injected single local bolus of lignocaine for alleviation of behavioural responses to pain during tail docking and castration of lambs with rubber rings. Research in Veterinary Science, 133, 210-218. doi:https://doi.org/10.1016/j.rvsc.2020.09.025
33) The Numnuts device injects the local anaesthetic product just at the exact moment as the ring compresses the tail, then the necessary time to anaesthetize the tissues before the acute vasoconstriction and pain begin has not elapsed.
An illustration of the device and local anaesthetic application has been added to the manuscript as figure 1. This diagram illustrates that the local anaesthetic is applied proximal to the ring (lamb side), although there is a delay in the application of LA and period of pain relief, lignocaine is a known fast acting agent and pain reduction is seen within the first 5 to 10 minutes post ring application.
Small, A. H., Jongman, E. C., Niemeyer, D., Lee, C., & Colditz, I. G. (2020). Efficacy of precisely injected single local bolus of lignocaine for alleviation of behavioural responses to pain during tail docking and castration of lambs with rubber rings. Research in Veterinary Science, 133, 210-218. doi:https://doi.org/10.1016/j.rvsc.2020.09.025
34) In addition, a high weight difference is showed between studied lambs. However, it is always used the same amount of product, so that amount may not cover all animals in the same way.
Local anesthetics are not administered on a per kg basis, rather the dose is chosen on the basis of the area being anaesthetized, hence the one dose can be appropriate for a range of lamb weights.
35) It is also observed that the number of observations for each lamb varied in hour 2 and 3, which can indicate a non-adequate capturing of the behavioural data. Moreover, teat seeking and sucking behaviour, which are shown decreased in Ring LA lambs, can indicate that the chronic pain produced is not well controlled during the trial. The residual discomfort that may have remained in the Ring LA lamb should have been considered in more detail to assess if this discomfort can produce problems in dairy growth or lead to long-term stress, which could finally influence the increase of secondary infections.
It is possible that behavioural observation may have been inadequate due to some animals having data points missing. However, the method in this study used to measure pain in lambs in response to ring castration and tail docking has been well validated. As lignocaine was used in this study only short term pain relief was expected. It is known that the best practice of provision of pain relief is through the use of multi modal analgesia (local anesthetics and nonsteroidal anti-inflammatory). As mentioned in response to Q31. ring tail docking has not been reported to have long term effects on lamb’s growth rate.
36) Finally, I would like to congratulate the authors on the high level of writing of the publication, and only a minimal correction was found in line 130. It should be written “There was” and not “The was”.
We thank the reviewer for taking the time to review the publication. L 130 has been edited.
Reviewer 3 Report
Dear authors,
In the article, it is not mentioned that the study has been approved by an ethical committee. Could the authors clarify this fact? An ethical committee must evaluate this type of studies in which harm is infringed on to animals.
In this trial, only acute pain, which is necessary to control, has been assessed. However, the evaluation of pain responses is challenging, especially in lambs, as they are very stoic animals and do not demonstrate pain as clearly and intensely as other species. This trial only assesses pain during the first three hours after treatment, and it is based on observational measurements such as abnormal behaviors or abnormal postures. This could have been improved with other measurements as serum cortisol or acute-phase proteins. The trial should have also included an assessment of the chronic pain that this type of procedure can generate in the animals due to the inflammation, trauma, stress and secondary infections after tissue necrosis.
The use of rubber rings to perform tail docking and castrations in animals is being banned in more and more countries due to the damage and stress generated to the animal. Although the main objective of this trial is to demonstrate that local anaesthetic can provide pain relief to avoid this damage, it is not considered during the trial the chronic pain, which an anti-inflammatory product could have controlled.
The Numnuts device injects the local anaesthetic product just at the exact moment as the ring compresses the tail, then the necessary time to anaesthetize the tissues before the acute vasoconstriction and pain begin has not elapsed.
In addition, a high weight difference is showed between studied lambs. However, it is always used the same amount of product, so that amount may not cover all animals in the same way.
It is also observed that the number of observations for each lamb varied in hour 2 and 3, which can indicate a non-adequate capturing of the behavioural data. Moreover, teat seeking and sucking behaviour, which are shown decreased in Ring LA lambs, can indicate that the chronic pain produced is not well controlled during the trial. The residual discomfort that may have remained in the Ring LA lamb should have been considered in more detail to assess if this discomfort can produce problems in dairy growth or lead to long-term stress, which could finally influence the increase of secondary infections.
Finally, I would like to congratulate the authors on the high level of writing of the publication, and only a minimal correction was found in line 130. It should be written “There was” and not “The was”.
Yours sincerely.
Author Response
Thank you for submitting a well-designed and well-written manuscript. I have only two comments:
- How did your reach a total of 30 animals (10 animals per group)? Did you perform an a priori power analysis to calculate the sample size?
As part of ethics application a priori power analysis was conducted. Power analysis of data from previous research provided the following indicative group sizes needed to observe effects of various magnitudes on the behaviors examined in this study.
Group size needed to have a 95% chance of detecting a significant effect (P < 0.05) of LA treatment on responses to ring castration
|
If reduction induced by LA treatment is |
Abnormal Postures first hour |
Pain avoidance behaviours first hour |
Abnormal postures over 3 hours |
|
30% |
5 |
31 |
8 |
|
50% |
2 |
9 |
4 |
|
70% |
1 |
4 |
1 |
- An image or graphical representation of the Numnuts® device would be helpful for the readers of the journal.
An image has been included in the manuscript as figure 1.
Reviewer 4 Report
Dear Authors,
Thank you for submitting a well-designed and well-written manuscript. I have only two comments:
- How did your reach a total of 30 animals (10 animals per group)? Did you perform an a priori power analysis to calculate the sample size?
- An image or graphical representation of the Numnuts® device would be helpful for the readers of the journal.
Author Response

(The authors gave the same response as above.)

Round 2
Reviewer 3 Report
Dear authors,
I would like to congratulate the authors on the high level of writing of the publication. Only minimal corrections have to been done.
Line 165 - the the two times period.
Line 190 - double space bar after There was an.
Line 193 - double space bar after There was no.
Line 195 - Word thehe
Line 199 - double space bar after … teat seeking behavior.
Thank you for indicating the approval of the ethics committee, as well as including information regarding chronic pain.
I still consider that it is needed the evaluation of pain responses when manipulating animals and harm is infrigted to them. t is obvious that acute pain control is essential and necessary. However, veterinarians should consider bot acute and chronic pain control. If you are only taking into acount the control of acute pain it looks like if you are trying to convince the farmer to buy your product. But like veterinarias and scientist, we must guarantee the welfare of the animal at all times, so we must take into account the long-term impact.
This aspect, together with the unequal distribution of the animals in the pens, the difference in their weight, and the taking of disparate measurements between the animals in the video recordings, make me doubt that the study design was suitable. Improving this conceptualization can greatly enhance the scientific interest of this test.
Finally, I think it is necessary to mention two aspects related to the bibliography. Firstable, The bibliography is very outdated. Most of the references are prior to the year 2000. Animal welfare is a current issue and more current references are available to complement them. It has been also noticed that most of the references do not show the names of all the authors, especially those included after the review. I consider it appropriate that all the authors are named in References, or at least that all the referenced bibliography is expressed in the same way.
Yours sincerely.